# Progressive Coordinate Transforms for Monocular 3D Object Detection

**Li Wang**[1][*]   **Li Zhang**[1][†]   **Yi Zhu**[2]   **Zhi Zhang**[2]   **Tong He**[2]   **Mu Li**[2]   **Xiangyang Xue**[1]

[1]Fudan University   [2]Amazon Inc.

## Abstract

Recognizing and localizing objects in the 3D space is a crucial ability for an AI agent to perceive its surrounding environment. While significant progress has been achieved with expensive LiDAR point clouds, it poses a great challenge for 3D object detection given only a monocular image. While there exist different alternatives for tackling this problem, it is found that they are either equipped with heavy networks to fuse RGB and depth information or empirically ineffective to process millions of pseudo-LiDAR points. With in-depth examination, we realize that these limitations are rooted in inaccurate object localization. In this paper, we propose a novel and lightweight approach, dubbed *Progressive Coordinate Transforms* (PCT) to facilitate learning coordinate representations. Specifically, a localization boosting mechanism with confidence-aware loss is introduced to progressively refine the localization prediction. In addition, semantic image representation is also exploited to compensate for the usage of patch proposals. Despite being lightweight and simple, our strategy leads to superior improvements on the KITTI and Waymo Open Dataset monocular 3D detection benchmarks. At the same time, our proposed PCT shows great generalization to most coordinate-based 3D detection frameworks. The code is available at: `https://github.com/amazon-research/progressive-coordinate-transforms`.

## 1 Introduction

Object detection is a fundamental and challenging task in scene understanding applications. Recently, 3D object detection has received increasing attention and found applications in a wide range of scenarios such as autonomous driving, robotics, visual navigation and mixed reality. Despite the great progress from the area of 2D object detection [34, 49, 40, 18, 4], 3D object detection remains a largely unsolved problem as it aims to predict the object location in the 3D space alongside 3D object dimension and orientation.

Existing prevalent approaches [50, 44, 35, 10, 11] for 3D object detection largely rely on LiDAR sensors, which provide accurate 3D point clouds of the scene. Although these approaches achieve superior performance, the dependence on expensive equipment severely limits their applicability to generic 3D perception. There also exists a cheaper alternative that takes a single-view RGB image as input, termed as monocular 3D object detection. However, its performance is far from satisfactory as itself is an ill-posed problem due to the loss of depth information in 2D image planes. Hence, several recent attempts introduce depth information to help monocular 3D detection. Such attempts can be roughly categorized into two directions, pixel-based and coordinate-based. Pixel-based

---

[*]Work done during an internship at Amazon.

[†]Li Zhang (lizhangfd@fudan.edu.cn) and Xiangyang Xue (xyxue@fudan.edu.cn) are the corresponding authors. Li Zhang is with School of Data Science, Fudan University. Li Wang and Xiangyang Xue are with School of Computer Science, Fudan University.

35th Conference on Neural Information Processing Systems (NeurIPS 2021).

Table 1: Probing investigation on coordinate-based methods, PatchNet [27] and Pseudo-LiDAR [42]. We examine the potential improvement by replacing the predicted factor with the corresponding ground truth. ∗ indicates our reproduced performance. We can see that coordinate-based methods mostly suffer from inaccurate localization.

| Factor | PatchNet* [$AP_{3D}/AP_{BEV}$] | | | Pseudo-LiDAR* [$AP_{3D}/AP_{BEV}$] | | |
| --- | --- | --- | --- | --- | --- | --- |
| | Mod. | Easy | Hard | Mod. | Easy | Hard |
| Baseline | 26.31/34.14 | 36.40/46.80 | 21.07/28.04 | 23.04/31.06 | 32.27/42.45 | 19.67/25.67 |
| dimension | 27.26/34.62 | 40.32/47.24 | 24.29/28.38 | 25.88/31.97 | 36.09/44.35 | 20.88/26.60 |
| rotation | 26.25/34.04 | 36.09/46.25 | 23.49/27.99 | 23.88/31.31 | 32.42/42.74 | 19.85/26.07 |
| x | 32.80/41.43 | 45.60/56.22 | 27.38/34.63 | 28.36/36.77 | 39.69/50.78 | 25.08/29.92 |
| y | 30.16/34.14 | 40.94/46.80 | 24.58/28.04 | 25.53/31.06 | 35.19/42.45 | 20.69/25.67 |
| z | 42.42/53.48 | 55.42/68.29 | 35.54/45.60 | 38.37/50.81 | 50.04/63.96 | 32.24/43.32 |
| location(xyz) | **72.58/75.27** | **81.41/85.14** | **57.69/66.10** | **64.36/73.37** | **79.13/83.77** | **55.64/58.27** |

approaches [9, 36, 31, 41] turn to use estimated depth map as additional input for improved detection performance. But at the same time, this leads to heavy computational burden and large memory footprint since they often operate on the entire image. Coordinated-based approaches [42, 29, 46, 27] pursue the coordinate representations as in LiDAR-based methods. They use the predicted depth map to convert the monocular image pixels to 3D coordinate representations, then apply a 3D detector on the converted coordinates. In particular, they are often lightweight since their network inputs are object proposals generated by 2D detectors [29, 27]. However, the performance of coordinated-based methods lags far behind LiDAR-based methods. So we ask, can we identify the bottleneck that holds back the 3D detection accuracy of coordinate-based methods and how can we improve them?

In order to determine the bottleneck, we conduct an investigation on two widely adopted coordinate-based methods, PatchNet [27] and Pseudo-LiDAR [42]. Specifically, for each prediction target, we examine the potential improvement by replacing its value with the corresponding ground truth, and then re-compute the 3D detection accuracy. As shown in Table 1, using ground truth dimension and rotation do not bring significant improvements over the baseline. But using ground truth location (*i.e.*, x/y/z values of the objects) almost triples detection accuracy. This indicates that coordinate-based methods mostly suffer from inaccurate localization even with the assistance of estimated depth maps.

Based on this observation, we focus on improving the accuracy of 3D center localization. In this work, we propose a lightweight and generalized approach, called Progressive Coordinate Transforms (PCT), to enhance the localization capability for coordinate-based methods. First of all, since the localization regression network in most coordinate-based methods is less accurate but lightweight, we propose to progressively refine its prediction similar to gradient boosting [12, 13]. To be specific, a localization regression network can be seen as a weak learner, and we progressively train multiple consecutive networks such that each network fits the regression residual from the previous networks. These networks share the same lightweight structure so that the computation overhead is negligible. We also predict a confidence score for each network to help stabilize the end-to-end training. We term this progressive refining strategy as confidence-aware localization boosting (CLB). Compared to image-only or pixel-based methods, coordinated-based methods suffer from the problem of missing global context information due to the use of patched input. In order to further improve the localization accuracy, we exploit semantic image representations from 2D detector. We term this module as global context encoding (GCE). We find that GCE can not only improve center localization accuracy, but also contribute to the final 3D box estimation.

Through extensive experiments, our progressive coordinate transforms, consisting of CLB and GCE, is shown to improve popular coordinate-based models [42, 27] by generating more accurate localization. Without bells and whistles, we achieve state-of-the-art monocular 3D detection performance on KITTI [16, 17, 15] with a strong base method [27]. Additionally, this also leads to superior improvements on Waymo Open Dataset [38] compared with the base method PatchNet.

## 2 Related work

### 2.1 Monocular 3D object detection

Existing paradigms for monocular 3D object detection can be categorized into two types: image-only methods and depth-assisted methods.

For image-only methods, they often adapt architectures and good practices from popular 2D detectors [34, 49, 40]. However, locating objects in 3D space is much more challenging without depth information. Hence, several works [30, 2, 25, 6, 49] integrate geometry consistency into the training strategy to constrain the localization prediction. Deep3DBox [30] divides orientation into multi-bins to stably regress them, and combines the 2D-3D box constraint to recover accurate 3D object pose. M3D-RPN [2] utilizes the geometric relationship between 2D and 3D perspectives by sharing the prior anchors and classification targets. MonoPair [6] leverages the spatial relationships between paired objects to improve accuracy on occluded objects. To further improve the performance of truncated objects, MonoFlex [48] decouples the features learning and prediction of truncated objects, and formulates an depth estimation to adaptively combine independent estimator based on uncertainty. [33] designs CaDDN as a fully differentiable end-to-end approach for joint depth estimation and object detection.

Depth-assisted methods often estimate a depth map given a input image, and use it in different ways. Some pixel-based approaches [9, 26] directly feed images and estimated depth maps into networks to generate depth-aware features and enhance the 3D detection performance. Some other coordinate-based approaches first transform the pixels of input images to 3D coordinates by leveraging the depth and camera information, then feed the coordinate proposals to a 3D detector. Pioneering work Pseudo-LiDAR [42] imitates the process of LiDAR-based approaches, which uses LiDAR-based 3D detector upon coordinates proposals. AM3D [29] explores the multi-modal input fusion to embed the complementary RGB cue into the network. Recently, PatchNet [27] points out that the efficacy of pseudo-LiDAR representation comes from the coordinate transform, instead of sophisticated LiDAR-based networks. Hence, they design a simple 2D CNN to perform 3D detection. In this work, we follow the research of coordinate-based methods [42, 27]. Instead of regressing 3D localization directly with a single lightweight network, we propose to progressively refine the prediction inspired by gradient boosting. We also incorporate RGB image information to complement patch proposals and enhance global context modeling. Different from AM3D [29], we utilize the RGB features from the 2D detector directly which can share the same context, and we do not need to train another RGB network from scratch.

## 2.2 Gradient boosting

Gradient boosting is a well-known greedy algorithm proposed in [7], which trains a sequence of learners and progressively improves the prediction results. It is a general learning framework, and has been verified to be a formidable force when applied with lightweight learners. Meanwhile, when each learner in the sequence is heavy, the computation cost becomes high and the performance is not beneficial [24]. Early works in 2D detection area [19, 21, 22] also adopt the boosting mechanism following a standard cascade paradigm, and achieve improved performance. Li et al. [21] treat face detection as an image retrieval task and improve it with a boosted exemplar-based face detector. Karianakis et al. [19] and Li et al. [22] feed convolutional features of proposals instead of hand-crafted features to boosted classifiers and distinguish objects from backgrounds. We can also find the usage of gradient boosting in other computer vision tasks [37, 51].

To our best knowledge, we are the first to explore the boosting mechanism in coordinate-based methods for 3D object detection. We perform this mechanism in two folds. First, instead of the entire 3D detection pipeline, we only progressively boost the localization regression network as its computational cost is insignificant comparing to the entire pipeline. Second, we refine the localization with an additional confidence score in the boosting procedure, such that the loss is balanced. These choices greatly improves the performance with small extra parameters.

## 3  Background

Before diving into the details, we first revisit recent coordinate-based monocular 3D detection methods and present a visual depiction of its common pipeline in Figure 1. The framework usually consists of four main components: 2D bounding box generation, depth map estimation, data transformation and 3D box estimation. Specifically, given an image $I$, the process can be described as:

*2D bounding box generation.* An off-the-shelf 2D object detector $F_{2d}$ such as Faster R-CNN [34] is employed on image $I$ to generate region of interests (RoIs), $\mathcal{R} = F_{2d}(I)$.

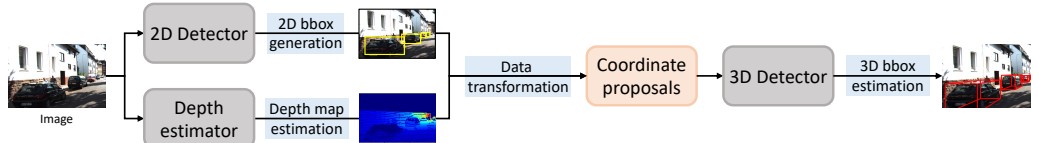

Figure 1: A common pipeline of coordinate-based monocular 3D detectors. It consists of four steps to predict the final 3D boxes: 2D bounding box generation, depth map estimation, data transformation and 3D box estimation. In this work, we focus on improving the last step: 3D box estimation.

*Depth map estimation.* An off-the-shelf depth estimator $F_z$ such as DORN [14] is applied on image $I$ to predict its depth map, $\mathcal{Z} = F_z(I)$.

*Data transformation.* To convert a pixel $(u, v)$ within a RoI to 3D space, the associated depth $z = \mathcal{Z}(u, v)$ is used to transform it into its 3D coordinates $(c_x, c_y, c_z)$ by

$$c_x = \frac{(u - u') \times z}{f_u}; \quad c_y = \frac{(v - v') \times z}{f_v}; \quad c_z = z \tag{1}$$

Here, $(c_x, c_y, c_z)$ is a pixel in the generated 3D coordinate patch $c$. $(u', v')$ is the camera principal point. $f_u$ and $f_v$ are the focal length along horizontal and vertical axis, respectively. $u', v', f_u, f_v$ are usually provided by the datasets.

*3D box estimation.* Once the 3D coordinates for each RoI are available, the final step is to predict 3D boxes with their center location, rotation and dimension. Different networks $F_{3d}$ such as Frustum PointNet [32] can be employed to conduct 3D box prediction, $\mathcal{B} = F_{3d}(c)$. Here, $\mathcal{B}$ includes the center location $(x, y, z)$, rotation $(\theta)$, and dimensions $(w, h, l)$ of the 3D box.

Since the first two steps use off-the-shelf models and the third step can be computed analytically, in this paper, we focus on improving the last step of coordinated-based methods. In particular, our goal is to improve the accuracy of localization prediction motivated by the observation in Table 1.

## 4 Method

In this section, we present our progressive coordinate transforms (PCT) for improved 3D detection. In order to obtain more accurate localization predictions, we introduce a confidence-aware localization boosting mechanism (CLB) in Sec. 4.1 to progressively refine the prediction. Then in Sec. 4.2, we incorporate RGB image information by a global context encoding (GCE) strategy to compensate for the drawbacks of using patch proposals. In the end, we illustrate the overall framework of PCT in Figure 2 (a).

### 4.1 Confidence-aware localization boosting

Following Frustum PointNet [32], most coordinate-based methods [42, 45, 27, 43] divide the last step of 3D box estimation into two major components. The first component is a lightweight 3D localization regressor $F$, whose input is 3D coordinate proposals generated from data transformation. The second component is a relatively heavy network $G$ used to regress the final 3D box $\mathcal{B}$. Recalling the results in Table 1, 3D localization performance is the weakest point of a coordinate-based model, accounting for up to 50 AP loss when all other modules keep intact. Therefore, can we find an efficient way to improve the accuracy of localization prediction and also generalizes to other coordinated-based methods?

Gradient boosting [12, 13] is a general learning framework that combines multiple weak learners into a single strong one in an iterative fashion. Let $\mathcal{L}(x)$ be the risk of ensemble models, the algorithm devotes to seek an approximation $h(x)$ to minimize $\mathcal{L}(y^*, x) = \Psi(y^*, h(x))$, where $y^*$ is a target value, $\Psi(\cdot)$ is the loss function. $h(x)$ is a linear combination of a set of weak (base) learners $f_t(x)$ from some class $\mathcal{F}$, *i.e.*, $h(x) = \sum_{t=1}^{t=T} \gamma_t f_t(x) + const$. Here, $T$ is the total training iterations and $\gamma_t$ is the corresponding weight for each weak learner. To minimize the empirical risk, the algorithm starts with a model $h_0(x)$, and then incrementally expands it in a greedy manner. This process manages to fit a new weak learner to the residual errors made by the previous set of learners. Mathematically, the

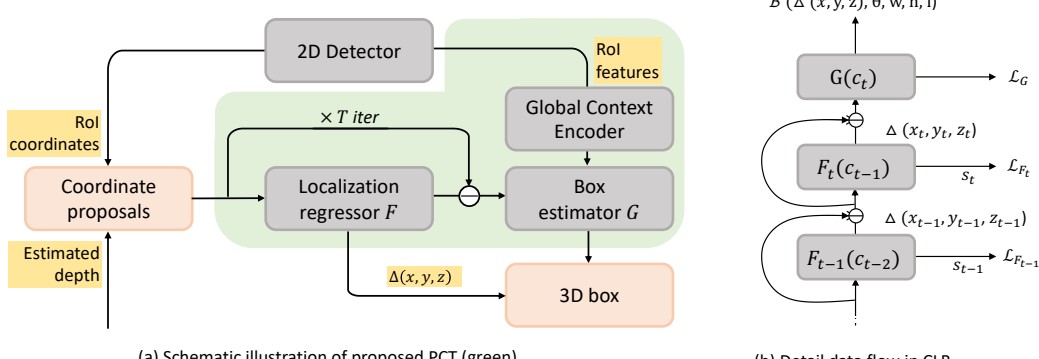

(a) Schematic illustration of proposed PCT (green)    (b) Detail data flow in CLB.

Figure 2: (a) Schematic illustration of proposed Coordinate Transforms (PCT). We treat $F$ as a weaker learner, and perform coordinate transform $T$ steps via confidence-aware localization boosting (CLB module) to obtain a better localization. Then the refined coordinate proposals combined with corresponding encoded RGB features (GCE module) are fed into network G to generate final 3D bounding boxes. (b) The data flow of CLB mechanism in detail. For step $t$, it takes refined coordinate patches $c_{t-1}$ as input, which is transformed based on predicted $\Delta(x_{t-1}, y_{t-1}, z_{t-1})$. Then network $F_t$ generate the residual localization for the next step, and confidence $s_t$ is also generated during the training process.

optimization can be formulated as

$$h_0(x) = \arg\min_{\gamma_0} \mathcal{L}_0(y^*, x);$$

$$\dots \tag{2}$$

$$h_t(x) = h_{t-1}(x) + \arg\min_{f_t \in \mathcal{F}} \mathcal{L}_t(y^*, h_{t-1}(x) + f_t(x)).$$

Inspired by gradient boosting, we imitate its optimization procedure to progressively adapt localization prediction by multiple localization regressors instead of a single one used in previous works [42, 29, 46, 27]. To be specific, we treat localization network $F$ as a weak learner and stack multiple of them as shown in Figure 2 (b). Given $F$ is a lightweight network, the extra computational cost brought by gradient boosting is insignificant. After the data transformation step, each 2D bounding box obtains its corresponding coordinate patch $c$. We take the coordinate patch as the input of weak learner $F$ to regress the center localization residual $\Delta(x, y, z)$ based on the prediction from previous stage,

$$\Delta(x_t, y_t, z_t) = F_t(c_{t-1}); \text{ where } c_{t-1} = c_0 - \gamma_0(x_0, y_0, z_0) - \sum_{i=1}^{t-1} \gamma_i \Delta(x_i, y_i, z_i). \tag{3}$$

We denote $c_0$ and $(x_0, y_0, z_0)$ to be the initial coordinate input patch and object location, respectively. Coordinate input patch $c_{t-1}$ is then transformed according to the localization residual prediction, and fed into the next weak learner.

Thus the risk at stage $t$ can be written as,

$$\mathcal{L}_{F_t}((x^*, y^*, z^*), c_{t-1}) = \Psi((x^*, y^*, z^*), \gamma_0(x_0, y_0, z_0) + \sum_{i=1}^{t-1} \gamma_i \Delta(x_i, y_i, z_i)) \tag{4}$$

where $(x^*, y^*, z^*)$ indicates the ground truth location. At this point, we can easily see that Eq 4 is a natural derivation from Eq 2. After $T$ iterations, the final adjusted prediction $c_T$ is fed into the network G to estimate the 3D box $\mathcal{B}$, $i.e. \mathcal{B} = G(c_T)$.

**Confidence-aware network loss** In the case that the target of weak learner $f$ is differentiable, gradient boosting solves the optimization problem in a forward greedy manner as shown in Eq 2. For each iteration, it first fits the weak learner $f$ to the residual error, and then the optimal value of the coefficient weight $\gamma$ is determined for this weak learner. The optimization procedures train iteratively for $T$ iterations.

However, for our center regression task, we would like to train $T$ weak localization networks $F_t(c_{t-1}), t \in 1,...,T$ and a 3D box prediction network $G$ in an end-to-end manner instead of bootstrapping. This is challenging in terms of both computational cost and optimization stability, given the simultaneous training of a set of weak learners and their coefficient weights. Therefore, we first simplify the problem by treating all $\gamma_t$ the same and only focus on optimizing the localization networks. However, the contribution from each weak learner $F$ may not be the same during end-to-end training, which leads to unstable optimization. Hence, we tailor a confidence-aware boosting loss to facilitate network training, by learning confidence score $s_t$ for each localization loss function $\mathcal{L}_{F_t}$. The confidence score $s_t$ is learned from a small decoder and followed a self-balancing formulation closely coupled to the network loss. The overall loss function is defined as

$$\mathcal{L}(\mathcal{B}^*, c_0) = \sum_{t=1}^{T} s_t * \mathcal{L}_{F_t}((x^*, y^*, z^*), c_{t-1}) + \lambda_s \prod_{t=1}^{T}(1 - s_t) + \mathcal{L}_G(\mathcal{B}^*, c_T), \qquad (5)$$

where $\mathcal{B}^*$ is the 3D box ground truth, $(x^*, y^*, z^*) \in \mathcal{B}^*$ and $\lambda_s$ is the balance weight. $s_t$ is the prediction after sigmoid, which represents the confidence of the localization regression loss at $t^{th}$ stage, and $\prod_{t=1}^{T}(1 - s_t)$ is the penalty on the network uncertainty. In other words, if $s_t$ is approaching 1, which means the network is confident about localization refinement at $t^{th}$ stage, then no penalty will be applied. Otherwise, the uncertainty of regression loss is high, thus triggers a higher penalty.

## 4.2 Global context encoding

Typical coordinate-based methods [42, 45, 27, 43] perform 3D detection based on 2D RoIs, which is similar to two-stage 2D detection frameworks, such as Faster R-CNN [34]. In a two-stage 2D object detection framework, the second stage reuses the features from the first stage via RoIPooling [34] or RoIAlign operators [18] guided by ROI proposals, and then a small decoder is used for localization refinement. However, in 3D detection, only cropped patches with coordinates information are fed to the network for 3D box regression. Neither RGB information nor global context is included.

Considering that the RGB information is a vital visual clue, we explore its aggregation in the last 3D box estimation step. Similar to two-stage 2D detectors, we obtain the RGB information by directly cropping the corresponding features from a 2D detector $F_{2d}$. Then the input to 3D box estimator $G$ can be formulated as $c = \{[\mathcal{D}(u, v), \mathcal{A}(u, v)], \forall (u, v) \in \mathcal{R}\}$. Here, $\mathcal{D}(\cdot)$ represents the data transformation function and $\mathcal{A}(\cdot)$ represents the RoIAlign operation. Both operations are performed upon the generated regions of interest from $\mathcal{R}$.

After RoIAlign operation, features are of size $C \times K \times K$, where $C$ is the number of channels and $K \times K$ is the corresponding width and height, respectively. A small feature encoder is then used to encode cropped features into vectors with the dimension of $C$. A feature fusion is followed to integrate coordinate representations with the obtained image representations. Benefiting from the large receptive fields of image feature representations, 3D box estimator can now have access to global context. Besides, directly cropping on RGB features also avoids learning image representations of RoIs from scratch and reduces the overall network parameters.

## 5 Experiments

Two monocular 3D detection benchmarks are introduced in Sec. 5.1 and Sec. 5.2, while experimental implementation details are described in Sec. 5.3. In Sec. 5.4, we conduct main analysis on KITTI dataset [16, 17, 15] with base method PatchNet [27] given its current best performance. More experiments on Waymo Open Dataset [38] are also demonstrated to further verify the generality of our proposed PCT in Sec. 5.5.

## 5.1 KITTI setup

We first evaluate our method on the KITTI benchmark [16, 17, 15], which contains 7,481 and 7,518 images for training and testing respectively. We follow [5] to split the 7,481 training images into 3712 for training and 3,769 for validation.

Table 2: Ablative analysis on KITTI validation set for $\mathrm{AP_{3D}}$ and $\mathrm{AP_{BEV}}$ at IoU = 0.7. Experiment Group (I) is the baseline method. Different experiment settings are explored: (II) applying localization boosting without confidence constraint, (III) performing confidence-aware localization boosting algorithm, (IV) adding global context encoding on Group (II), (V) our full approach.

| Group | Localization Boosting | Uncertainty | GCE | $\mathrm{AP_{3D}}$ | | | $\mathrm{AP_{BEV}}$ | | |
|-------|------------------------|-------------|-----|------|------|------|------|------|------|
| | | | | Mod. | Easy | Hard | Mod. | Easy | Hard |
| I | - | - | - | 25.88 | 36.07 | 20.99 | 33.34 | 46.39 | 27.54 |
| II | ✓ | - | - | 26.78 | 37.29 | 24.11 | 34.39 | 47.08 | 28.28 |
| III | ✓ | ✓ | - | 27.24 | 38.32 | 24.39 | 33.92 | 46.77 | 27.98 |
| IV | ✓ | - | ✓ | 27.12 | 37.38 | 24.11 | 34.46 | 46.70 | 28.32 |
| V | ✓ | ✓ | ✓ | **27.53** | **38.39** | **24.44** | **34.65** | **47.16** | **28.47** |

Precision-recall curves are adopted for evaluation, and we report the average precision (AP) results of 3D and Bird's eye view (BEV) object detection on KITTI validation and test set. For fair comparison to previous literature, the 40 recall positions-based metric $AP|_{R40}$ is reported on test set while $AP|_{R11}$ is reported on validation set. Three levels of difficulty are defined in the benchmark according to the 2D bounding box height, occlusion, and truncation degree, namely, "Easy", "Mod.", and "Hard". The KITTI benchmark ranks all methods based on the $\mathrm{AP_{3D}}$ of "Mod.". In particular, we focus on the "Car" category as in [42, 27], and we adopt IoU = 0.7 as threshold for evaluation.

## 5.2 Waymo setup

We also carry out experiments on large-scale, high quality and diverse dataset, Waymo open dataset [38]. It provides pre-defined 798 training sequences and 202 validation sequences from different scenes, and another 150 test sequences without labels. The dataset contains camera images from five high-resolution pinhole cameras, and we only consider images with their 3D labels from front camera for monocular 3D detection task. We sample every third frame from the training sequences (total 52,386 images) as in CaDDN [33] to form the training set due to its large scale. And validation set contains all the 39,848 images from 202 different scenes.

For evaluation, we adopt the officially released evaluation [39] to calculate the mean average precision (mAP) and the mean average precision weighted by heading (mAPH). Two levels are included according to difficulty rating, which are defined by LiDAR points. 3D labels without any points are ignored, LEVEL_2 is assigned to examples when it contains equal or lesser than 5 points, while the rest of the examples are assigned to LEVEL_1. Additionally, three distances (0 - 30m, 30 - 50m, 50m - ∞) to sensor are considered during evaluation.

## 5.3 Implementation details

Our overall framework of PCT can be visualized in Figure 2. In terms of implementation details, we instantiate our algorithm on two widely adopted coordinate-based methods with public released code [28], PatchNet and Pseudo LiDAR. Bearing efficiency in mind, we use a real-time 2D detector RTM3D [23] with DLA-34 [47] as backbone. For the sake of fair comparison, we adopt depth predictor DORN [14] on KITTI dataset as in most depth-assisted literature. Since there is no published depth results on Waymo open dataset, we adopt a most recent monocular depth estimator AdaBins [1] trained on Waymo training set. For the CLB mechanism, we inherit the original localization regression framework in each method. Each $F_t$ shares the same structure. The corresponding confidence is generated following the last convolutional layers of $F_t$ with three linear layers and a Sigmoid function. $T = 3$ and $\lambda_s = 1$ are set for the following experiments except for the ablation study on boosting iterations. For GCE module, we get the corresponding input image features by performing RoIAlign on the features from last convolutional layer of 2D detector. We set the output of RoIAlign as $16 \times 16$. As 2D detector use DLA-34 as backbone, the obtained image feature representations have the size of $64 \times 16 \times 16$ and entitle arbitrary sized receptive field theoretically due to the embedded deformable convolution [8]. The structure of feature encoder in global context module is two common $64 \times 3 \times 3$ convolutional layers (stride=4) and a $64 \times 1 \times 1$ convolutional layer (stride=1). Hence, image feature representations are encoded to a vector with 64-dim. The obtained features are then concatenated with the coordinate feature vectors from the final global pooling of box prediction network $G$. With the lightweight structure, we are able to optimize the network end-to-end on a single Nvidia V100

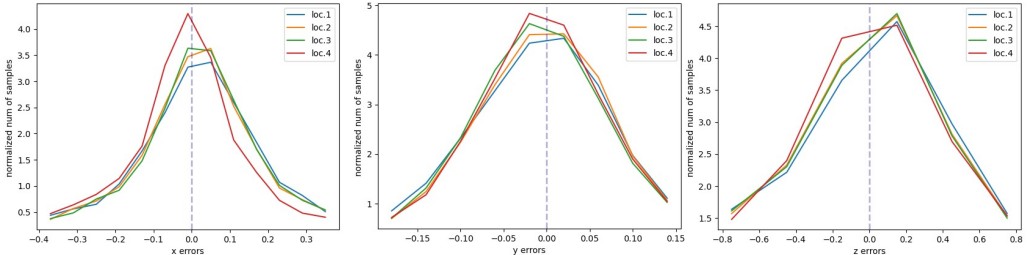

Figure 3: The statistic analysis and comparison on different Localization boosting stage when $T = 3$. The vertical axis of the chart represents the number of samples after normalization. "loc.1/2/3" denotes the $1/2/3^{th}$ step of localization errors in $F$ and "loc.4" is the final localization errors in $G$. Note that when the curve is more thin, tall, and closer to zeros, the localization is more accurate.

GPU with 16G memory. The training criterion for network $F$ and $G$ and other training settings follow the corresponding base methods for fair comparison.

## 5.4 Method analysis on KITTI dataset

**Main ablative analysis** In Table 2, we conduct ablation studies to analyze the effectiveness of our contributions: I) Without any localization regression network $F$, network $G$ generates 3D box prediction directly. II) This configuration only contains localization boosting part without confidence constraint. III) The entire CLB mechanism is included to progressively regress center localization. IV) GCE module is added to the network based on the localization boosting block without confidence constraint since the feature fusion can be performed on either the localization regression networks $F$ or 3D box prediction network $G$. V) Our full method with all the components.

As depicted in Table 2, we can observe that the performance continues to grow with the addition of every component. From group II, localization boosting brings a noticeable improvement on all settings especially "Hard" set, which confirms its effectiveness in increasing localization accuracy. Group III shows that balancing training loss by adding confidence leads to better and more stable optimization. Group IV reveals that the proposed GCE module can effectively equip RGB information and global context with 3D coordinate representations. In the end, Group V demonstrates the complementarity of the proposed CLB mechanism and GCE module, leading to an improvement from 25.88/36.07/20.99 to 27.53/38.39/24.44 compared with Group I.

Table 3: Comparison of different boosting iteration settings on KITTI validation split set.

| Localization | $AP_{3D}/AP_{BEV}$ | | |
| Boost ($T$) | Mod. | Easy | Hard |
| --- | --- | --- | --- |
| - | 25.88/33.34 | 36.07/46.39 | 20.99/27.54 |
| 1 | 26.31/34.14 | 36.40/46.80 | 21.07/28.04 |
| 2 | 26.69/34.06 | 37.17/46.42 | 24.04/28.03 |
| 3 | **26.78/34.39** | **37.29/47.08** | **24.11/28.28** |
| 4 | 26.77/34.21 | 37.12/47.00 | 23.48/28.23 |
| 5 | 26.64/34.43 | 37.24/47.04 | 23.89/28.27 |

Table 4: Evaluation of different coordinate feature fusion with GCE on KITTI validation set. Baseline is the Group (II) in Table 2.

| Method | $AP_{3D}$ | | |
| | Mod. | Easy | Hard |
| --- | --- | --- | --- |
| Baseline | 26.78 | 37.29 | 24.11 |
| $F$ + GCE | 27.08 | 37.33 | 24.07 |
| $G$ + GCE | **27.12** | 37.38 | 24.11 |
| All + GCE | 27.07 | **37.43** | **24.18** |

**Localization boosting iteration settings** We explore the effect of different localization boosting iteration settings in this part. For a fair comparison, we do not perform the confidence constraint on regression loss. As illustrated in Table 3, when boosting iteration $T = 3$, we achieve the best 3D detection performance. More iterations of boosting do not bring improvements, which might be caused by overfitting with the increasing of network parameters.

To verify the improvement of each step in boosting procedure, we conduct the comparison of localization errors at iteration $T = 3$ on the specific metrics (location "xyz") of the ground truth. In particular, three stacked localization networks $F$ generate intermediate localization "loc.1/2/3" and $G$ output the final localization "loc.4". As shown in Figure 3, we can see that the distributions of "x",

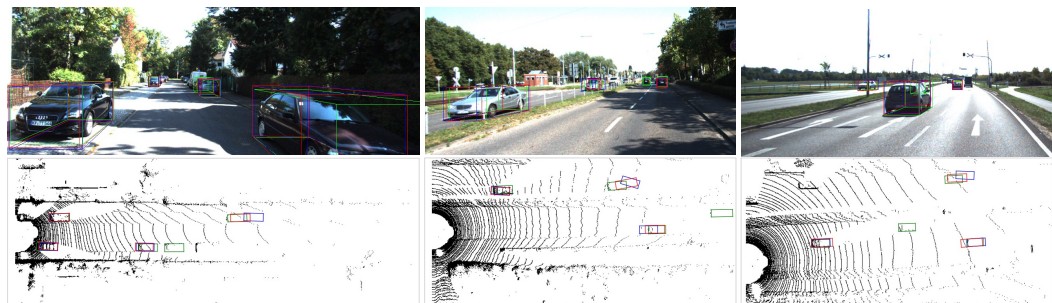

Figure 4: Qualitative comparison of ground truth (green), base method PatchNet (blue), and our method (red) on KITTI val set. The first and second rows show RGB and BEV images respectively.

Table 5: Comparison of generalization on KITTI validation set. ∗ denotes that the method is reproduced by ourselves.

| Method | AP$_{3D}$ | | | AP$_{BEV}$ | | |
|---|---|---|---|---|---|---|
| | Mod. | Easy | Hard | Mod. | Easy | Hard |
| Pseudo-LiDAR*[42] | 23.04 | 32.27 | 19.67 | 31.06 | 42.45 | 25.67 |
| Pseudo-LiDAR + CLB | 24.14 | 34.46 | 20.16 | 32.41 | 44.98 | 26.82 |
| Pseudo-LiDAR + CLB + GCE | 24.43 | 34.34 | 20.18 | 32.50 | 45.35 | 26.91 |

"y" and "z" errors tend to distributed to zero with localization boosting iterating. For instance, the red line in left chart is narrow and tall near zero along horizontal axis compared with other lines, which means that the corresponding x coordinate is more accurate than others. This further suggests that localization boosting is useful for object localization. The corresponding BEV visualization will be shown in Supplementary Material.

**Impact of global context encoding**   We also explore where the global context representation fusion operates. We take Group II as the baseline, and perform feature fusion on localization regression network $F$ (row one), 3D estimation network $G$ (row two) or on both (final row). RoI features are encoded into a vector with GCE and then concatenate with the feature vectors from network ($F$ or $G$) global pooling. As shown in Table 4, the operation on $G$ outperforms it on $F$, which indicates that image representation is more suitable for the overall box prediction rather than only localization as it contains the additional semantic appearance information. Although operation on all networks achieves a lightly higher than it on $G$ on the "Easy" and "Hard" set, introducing parameters is much larger due to operation on stacked localization networks. Hence, we only apply GCE on network $G$ in our approach for a lightweight network and avoid overfitting during training.

**Generally applicable to other coordinate-based algorithm**   In this section, we demonstrate the generalization capability of our algorithm to classic coordinate-based methods Pseudo-LiDAR [42]. As shown in Table 5, each component of our algorithm improves the original methods a lot. Specially, our approach improves Pseudo-LiDAR by 1.43/2.06/0.51 while 1.22/1.99/3.36 gains on PatchNet.

**Comparison with state-of-the-arts**   We build our test detector on the current state-of-the-art coordinate-based method PatchNet, and results are shown in Table 6. Quantitatively, our method achieves the highest performance on "Mod." set with 22 FPS on NvidiaTesla v100 including 2D detector inference time, which is the main setting for ranking on the benchmark. Specially, large margins, 2.25/5.32/1.14 on 3D detection and 2.17/6.68/0.95 on BEV, are observed over the base method PatchNet with only additional 3.41M parameters. Besides, our methods also outperforms the pixel-based state-of-the-arts methods Liu et al. [26] especially on "Hard" set.

Qualitative comparisons are shown in Figure 4. The ground truth, base method (PatchNet), and our method are colored in green, blue, and red, respectively. For better visualization, the first and second rows show RGB images and BEV images, respectively. Compared with the base method, our algorithm can produce higher-quality 3D bounding boxes in different kinds of scenes.

Table 6: Comparison with SoTA methods on the KITTI test set at IoU = 0.7. Our algorithm achieves new SoTA performance. "Depth" means if the method belongs to depth-assisted methods or not. "Type" indicates the method input pattern, "Pixel" denotes the methods with image as inputs directly while "Coordinate" means the coordinate-based methods with 3D coordinates as inputs.

| Method | Depth | Type | $AP_{3D}$ | | | $AP_{BEV}$ | | |
| --- | --- | --- | --- | --- | --- | --- | --- | --- |
| | | | Mod. | Easy | Hard | Mod. | Easy | Hard |
| AM3D [29] | yes | Coordinate | 10.74 | 16.50 | 9.52 | 17.32 | 25.30 | 14.91 |
| PatchNet [27] | yes | Coordinate | 11.12 | 15.68 | 10.17 | 16.86 | 22.97 | 14.97 |
| GrooMeD-NMS [20] | no | Pixel | 12.32 | 18.10 | 9.65 | 18.27 | 26.19 | 14.05 |
| Kinematic3D [3] | yes | Pixel | 12.72 | 19.07 | 9.17 | 17.52 | 26.69 | 13.10 |
| DDMP-3D [41] | yes | Pixel | 12.78 | 19.71 | 9.80 | 17.89 | 28.08 | 13.44 |
| Liu et al. [26] | yes | Pixel | 13.25 | **21.65** | 9.91 | 17.98 | **29.81** | 13.08 |
| **PCT** | yes | Coordinate | **13.37** | 21.00 | **11.31** | **19.03** | 29.65 | **15.92** |

## 5.5 Results on Waymo Open Dataset

Table 7 shows the results of base method PatchNet [27] and our proposed PCT. It can be observed that our method consistently outperforms the base method on mAP/mAPH of 0.50%/0.51% and 0.28%/0.30% on the LEVEL_1 and LEVEL_2 difficulties respectively under IoU = 0.7. Again, our method is efficient, e.g, it takes 5 days to complete training on large scale Waymo dataset with a 8-GPU node. More qualitative results can be seen at Supplementary Material.

Table 7: 3D performance on Waymo validation set. We demonstrate results of base method Patch-Net [27] and corresponding PCT at IoU = 0.7 and I0U = 0.5. Our proposed PCT achieves consistent improvements on all settings.

| Difficulty | Threshold | Method | 3D mAP / 3D mAPH | | | |
| --- | --- | --- | --- | --- | --- | --- |
| | | | Overall | 0 - 30m | 30 - 50m | 50 - $\infty$ |
| LEVEL_1 | IoU=0.7 | PatchNet | 0.39 / 0.37 | 1.67 / 1.63 | 0.13 / 0.12 | 0.03 / 0.03 |
| | | PCT | 0.89 / 0.88 | 3.18 / 3.15 | 0.27 / 0.27 | 0.07 / 0.07 |
| | IoU=0.5 | PatchNet | 2.92 / 2.74 | 10.03 / 9.75 | 1.09 / 0.96 | 0.23 / 0.18 |
| | | PCT | 4.20 / 4.15 | 14.70 / 14.54 | 1.78 / 1.75 | 0.39 / 0.39 |
| LEVEL_2 | IoU=0.7 | PatchNet | 0.38 / 0.36 | 1.67 / 1.63 | 0.13 / 0.11 | 0.03 / 0.03 |
| | | PCT | 0.66 / 0.66 | 3.18 / 3.15 | 0.27 / 0.26 | 0.07 / 0.07 |
| | IoU=0.5 | PatchNet | 2.42 / 2.28 | 10.01 / 9.73 | 1.07 / 0.94 | 0.22 / 0.16 |
| | | PCT | 4.03 / 3.99 | 14.67 / 14.51 | 1.74 / 1.71 | 0.36 / 0.35 |

## 6 Conclusions

In this paper, we have introduced a novel approach PCT to address the inaccurate localization problem for monocular 3D object detection. This is achieved by iteratively transforming the coordinate representation with a confidence-aware booting mechanism. Meanwhile, global context is introduced to compensate for the missing of semantic image representation in coordinated-based methods. Through extensive experiments, we have shown that our proposed PCT substantially improve the performance of the coordinate-based model by a large margin, and achieve state-of-the-art monocular 3D detection performance on KITTI test set. Moreover, we also show consistent improvements compared to the strong baseline on the large-scale Waymo Open dataset.

There are several limitations that could indicate the possible directions for future work. First, the performance of off-the-shelf 2D detector directly influences the accuracy of coordinate-based methods, hence how to effectively design the 3D box estimation algorithm to fit with existing 2D detectors is important. Second, we only concentrate on the lightweight coordinate-based methods. It requires further exploration to extend our approach to pixel-based methods. Finally, our proposed global context encoding is a simple module. Despite working well, a more tailored feature fusion strategy between coordinate representation and RGB image representation is worth exploring.

**Potential impacts.** Our method focuses on the monocular 3D detection which can be applied in autonomous driving field. One potential social problem of our work is that it may aggravate the employment crisis of human servants and drivers, which replaces human with autonomous robots and intelligent systems.

## Acknowledgments

This work was supported by Shanghai Municipal Science and Technology Major Projects (No.2021SHZDZX0103 and No.2018SHZDZX01).

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
