# Supplementary Material: Progressive Coordinate Transforms for Monocular 3D Object Detection

**Li Wang**[1]* **Li Zhang**[1]† **Yi Zhu**[2] **Zhi Zhang**[2] **Tong He**[2] **Mu Li**[2] **Xiangyang Xue**[1]

[1]Fudan University    [2]Amazon Inc.

## 1 Overview

In this supplementary material, we provide additional experimental results and qualitative visualizations. Specifically, we demonstrate the impacts of using different off-the-shelf models in Sec. 2, including 2D detectors and depth estimators. We show that our proposed PCT method achieves consistent improvements with all configurations. Additional qualitative results are visualized in Sec. 3. We present both successful predictions and failure cases. Our results suggest that these failure cases can often come from two aspects, low recall of the 2D detectors and the rotation errors in 3D box prediction.

## 2 Additional experiments

In this section, we firstly show the additional results on "Pedestrian" and "Cyclist" categories.Then, we also conduct complexity analysis to verify the lightweight property of our model. At last, we would like to demonstrate the impacts of using different 2D detectors and depth estimators on the performance of 3D detection, which is the first two steps of the coordinate-based methods as mentioned in Sec. 3 of the main submission. Following the main submission, we conduct the experiments on the KITTI [6, 7, 5] dataset.

**Results on "Pedestrian" and "Cyclist"**    Non-rigid structures and various shape make it more challenging for monocular 3D detection to accurately detect "Pedestrian" and "Cyclist". Most previous methods [12, 1, 8] fail to demonstrate these two categories results, however, we report results on these two categories in Table 1 to show the generalization of our PCT. Following [3], we demonstrate $AP_{11}$ 3D object detection results on KITTI validation set at IoU = 0.5. As illustrated in Table 1, our method still outperforms the base method PatchNet, benefiting from the more accurate localization and complementary global context information. Besides, we also achieves a better performance compared with state-of-the-art pixel-based methods [3, 11].

---

*Work done during an internship at Amazon.

†Li Zhang (lizhangfd@fudan.edu.cn) and Xiangyang Xue (xyxue@fudan.edu.cn) are the corresponding authors. Li Zhang is with School of Data Science, Fudan University. Li Wang and Xiangyang Xue are with School of Computer Science, Fudan University.

35th Conference on Neural Information Processing Systems (NeurIPS 2021).

Table 3: 3D detection performance on the KITTI validation set. We explore two 2D detectors, one is from RTM3D [8] and the other is from DDMP-3D [11]. PatchNet [9] is used as baseline, and ∗ denotes our reproduced version. We demonstrate that different 2D detectors will lead to drastically different 3D detection results, and the performance of 2D detectors is not positively related to the final 3D detection accuracy.

| Methods | RTM3D* [$AP_{3D}/AP_{BEV}$] | | | DDMP-3D* [$AP_{3D}/AP_{BEV}$] | | |
| --- | --- | --- | --- | --- | --- | --- |
| | Mod. | Easy | Hard | Mod. | Easy | Hard |
| PatchNet* [9] | 26.31/34.14 | 36.40/46.80 | 21.07/28.04 | 22.16/32.63 | 33.84/45.64 | 20.17/27.28 |
| PatchNet* + PCT | **27.53/34.65** | **38.39/47.16** | **24.44/28.47** | **25.90/33.70** | **37.00/46.45** | **23.57/27.96** |

Table 4: 3D detection performance on the KITTI validation set. We explore two depth estimators, DORN and PSMNet. We adopt two baselines, Pseudo-LiDAR [12] and PatchNet [9], ∗ denotes our reproduced version. We demonstrate that a better depth estimator will lead to better 3D detection performance. In addition, our proposed PCT is able to achieve consistent improvements.

| Methods | DORN [$AP_{3D}/AP_{BEV}$] | | | PSMNet [$AP_{3D}/AP_{BEV}$] | | |
| --- | --- | --- | --- | --- | --- | --- |
| | Mod. | Easy | Hard | Mod. | Easy | Hard |
| Pseudo-LiDAR* [12] | 23.04/31.06 | 32.27/42.45 | 19.67/25.67 | 42.01/52.63 | 58.27/70.91 | 34.99/44.61 |
| PatchNet* [9] | 26.31/34.14 | 36.40/46.80 | 21.07/28.04 | 47.30/56.59 | 68.88/74.87 | 39.13/47.80 |
| Pseudo-LiDAR* + PCT | 24.43/32.50 | 34.34/45.35 | 20.18/26.91 | 45.31/54.59 | 61.90/72.96 | 37.61/45.79 |
| PatchNet* + PCT | **27.53/34.65** | **38.39/47.16** | **24.44/28.47** | **48.27/57.11** | **70.73/80.65** | **39.97/48.14** |

Table 1: 3D object detection performance for "Pedestrian"/"Cyclist" on KITTI validation set at IoU = 0.5. * denotes that the method is reproduced by ourselves.

| Method | Cyclist / Pedestrian | | |
| --- | --- | --- | --- |
| | Mod. | Easy | Hard |
| D$^4$LCN [3] | 4.41 / 11.23 | 5.85 / 12.95 | 4.14 / 11.05 |
| DDMP-3D [11] | 6.47 / 12.11 | 4.18 / 14.42 | 6.27 / 12.05 |
| PatchNet *[12] | 11.60 / 12.17 | 13.76 / 14.55 | 11.37 / 12.00 |
| PCT | **12.28 / 15.31** | **15.98 / 17.19** | **12.19 / 13.12** |

Table 2: Comparison of different model parameters. Our PCT only introduces marginal parameters based on PatchNet.

| Method | Params |
| --- | --- |
| DDMP-3D [11] | 285.50M |
| CaDDN [10] | 191.24M |
| PatchNet | 48.39M |
| PatchNet + PCT | 51.80M |

**Complexity analysis**    In this part, we analyze the complexity of our proposed method. As shown in Table 2, our proposed PCT only introduces 3.41M extra parameters, which is marginal compared to the base method PatchNet with 48.39M parameters. This verifies that our proposed PCT is lightweight, but can achieve better performance than PatchNet.

Besides, we also compare ours to the model sizes of recent pixel-based methods, such as CaDDN [10] and DDMP-3D [11]. From the table, we can see that our final model (PatchNet + PCT) is far smaller than pixel-based methods (5x lighter) but achieves competitive performance, which demonstrates that coordinate-based methods are promising and effective.

**Impact of different 2D detectors.**    Recall in Sec. 6 of the main paper, we point out that different 2D detectors will directly influence the performance of coordinate-based methods. Here, we still adopt PatchNet [9] as the baseline, but take 2D detectors from RTM3D [8] ($AP_{2D}$: 83.69/90.66/67.53) and DDMP-3D [11] ($AP_{2D}$: 89.47/90.73/80.60) to generate the regions of interest. The rest steps remain the same to highlight the impact of using different 2D detectors.

As reported in Table 3, different 2D detectors will lead to a great performance gap. For instance, our method (last row) can gain 1.63/1.39/0.87 on 3D detection AP ($AP_{3D}$), switching from the 2D detector in DDMP-3D to the one in RTM3D. Interestingly, we find that the accuracy of the 3D detector is not positively related to the accuracy of the 2D detector. Hence, how to effectively design the coordinate-based monocular 3D detection algorithm considering the 2D detector in a unified framework should be explored further.

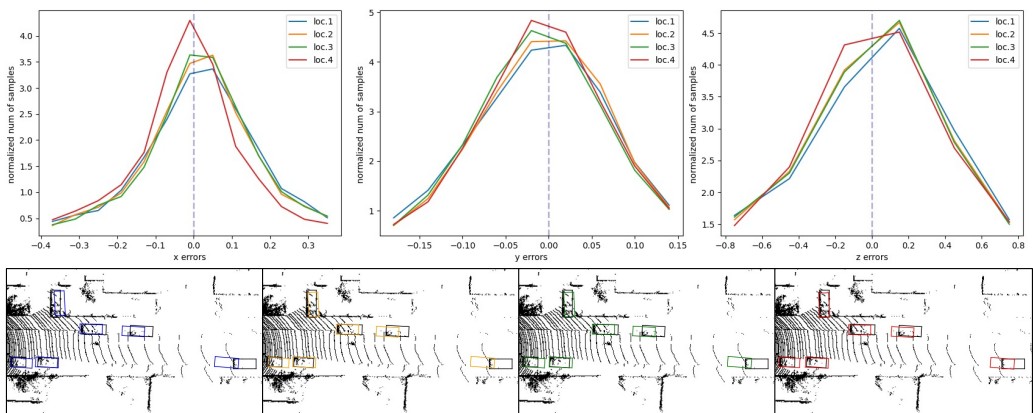

Figure 1: The statistic analysis and comparison on different Localization boosting stage when $T = 3$. The second row shows the BEV view and ground truth (black) at corresponding steps.

**Impact of different depth estimators.** We also explore the generalization of our method with respect to different depth estimators. Following depth-assisted methods [12, 4], we choose a stereo-based depth estimation method, PSMNet [2], to extract more accurate depth maps.

As illustrated in Table 4, we can see great performance improvement from using a better depth estimator. This supports our observation that coordinated-based methods mainly suffer from inaccurate localization. Most importantly, our proposed method PCT can still show obvious improvement even for a strong baseline.

## 3 Additional qualitative results

We firstly show the BEV prediction at each step in Figure 1, it can be seen that the predicted box gradually close to the ground truth box (shown in black), which conforms to the experiment results.

Figure 2 and Figure 3 shows more qualitative results on the KITTI dataset and Waymo open dataset. The 3D ground-truth boxes, and our method based on PatchNet [9] are drew in green and red, respectively. As clearly observed in Figure 2, our method can produce high-quality 3D bounding boxes in various scenarios with different lighting conditions and occlusions, and in various locations such as cities, residential districts and roads. Additionally, Figure 3 show the excellent results on large scale dataset, we demonstrate different time of Day in various scenarios: Day, Night, Dust and Dawn. Different weather conditions under daytimes are also shown, including sun, rain and fog.

Besides, we also illustrate some failure cases in different scenes on KITTI dataset to analyze the limitations. As shown in Figure 4 (a), the low recall rate of the 2D detector leads to the performance drop of 3D bounding boxes prediction. For example, the cars are occluded too heavily to be detected for a 2D detector in the second row of Figure 4 (a), thus the corresponding 3D prediction will not be performed. Meanwhile, the rotation deviations in Figure 4 (b) also indicate the appearance misperception. In the case of the last two rows in Figure 4 (b), the occlusion of cars leads to a great deviation on car rotation prediction. Hence, we believe that off-the-shelf 2D detectors and appearance information are important factors to consider in future work when designing a 3D detector.

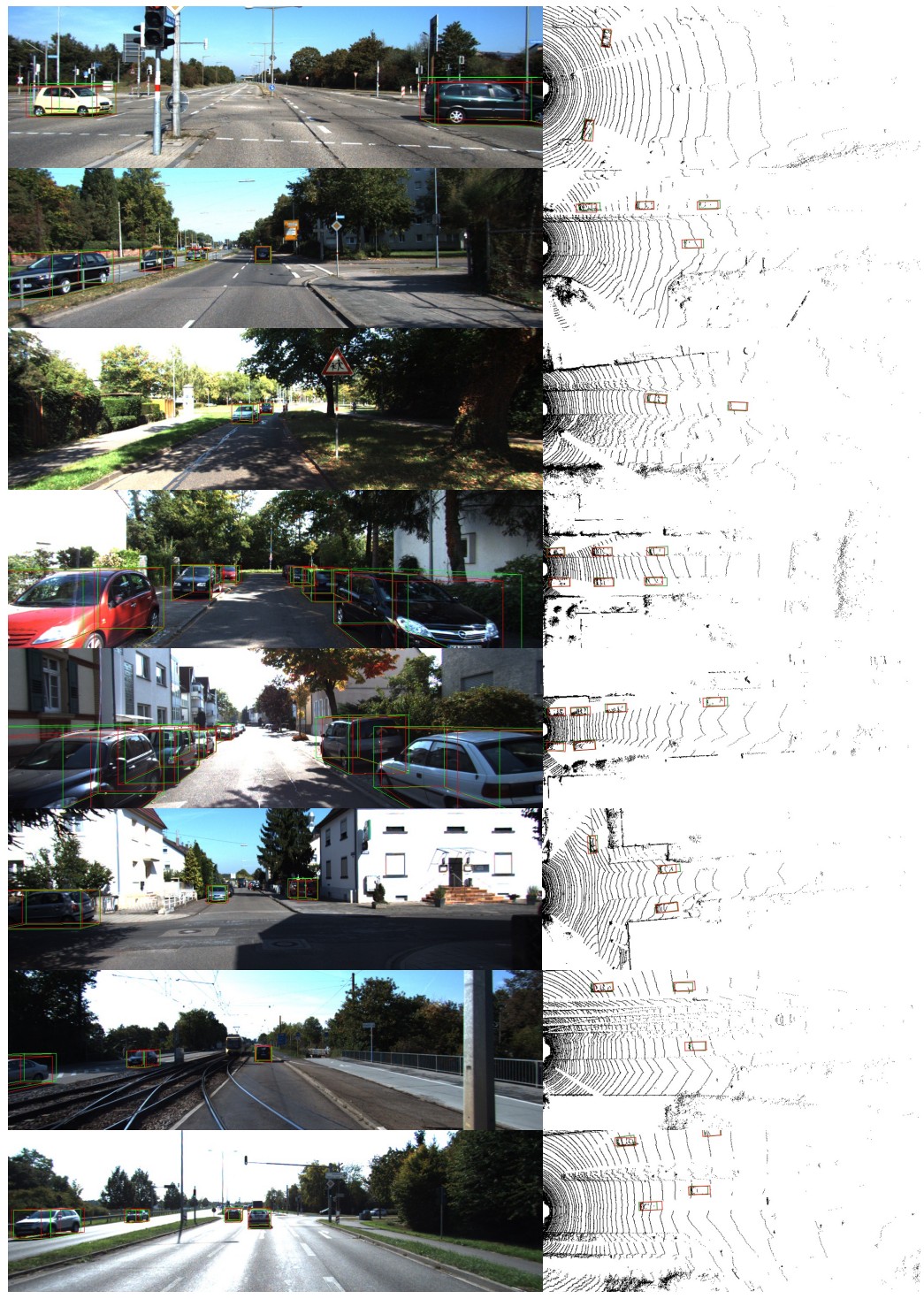

Figure 2: More qualitative results on the KITTI validation dataset. The 3D ground-truth boxes and our predictions are drew in green and red, respectively. We demonstrate the results in various scenarios with different lighting conditions and occlusions, including cities, residential districts and roads.

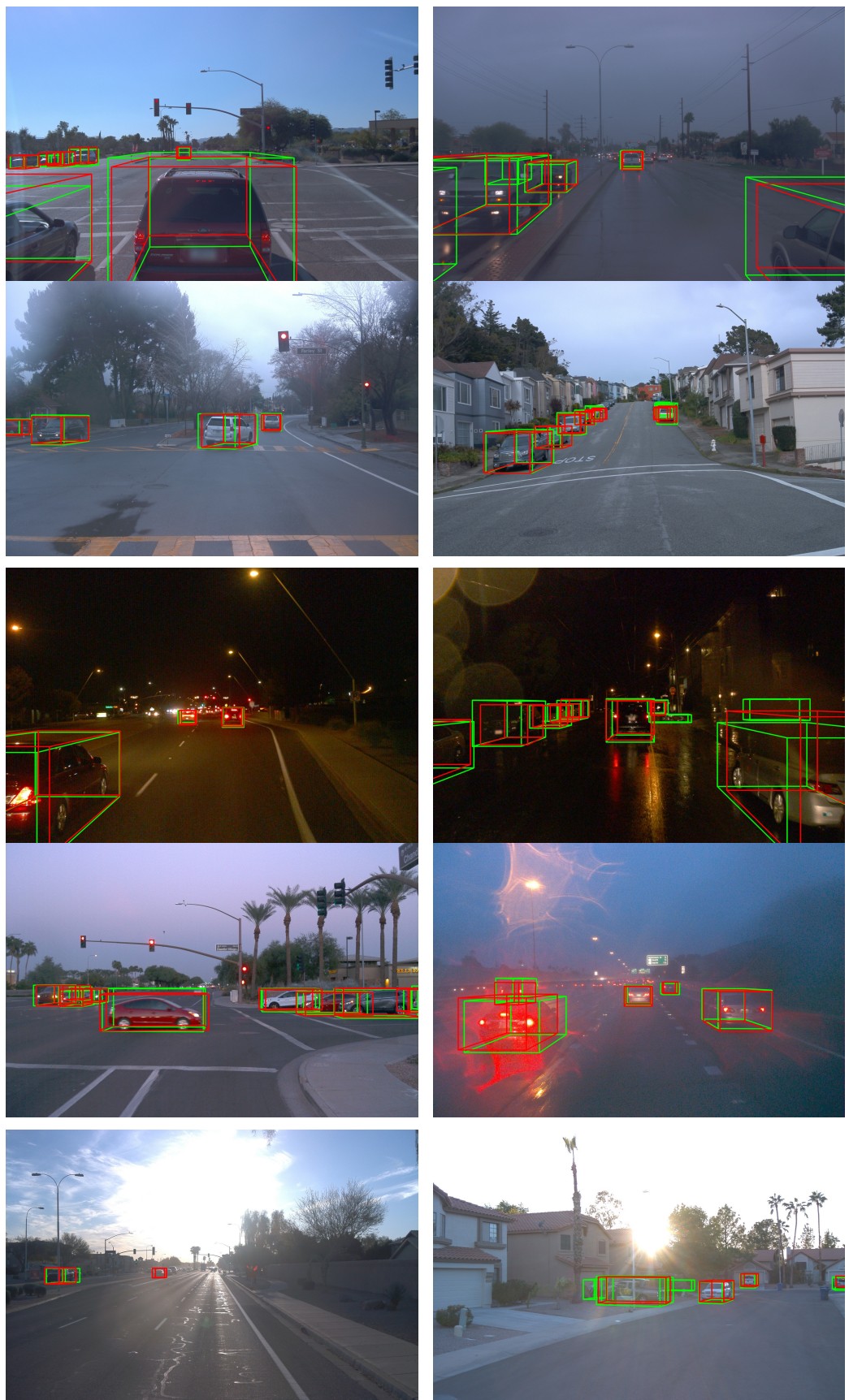

Figure 3: More qualitative results on the Waymo validation set with different time of day: Day, Night, Dust, Dawn and different weather: sun, rain and fog. The ground-truth and our predictions are drew in green and red, respectively.

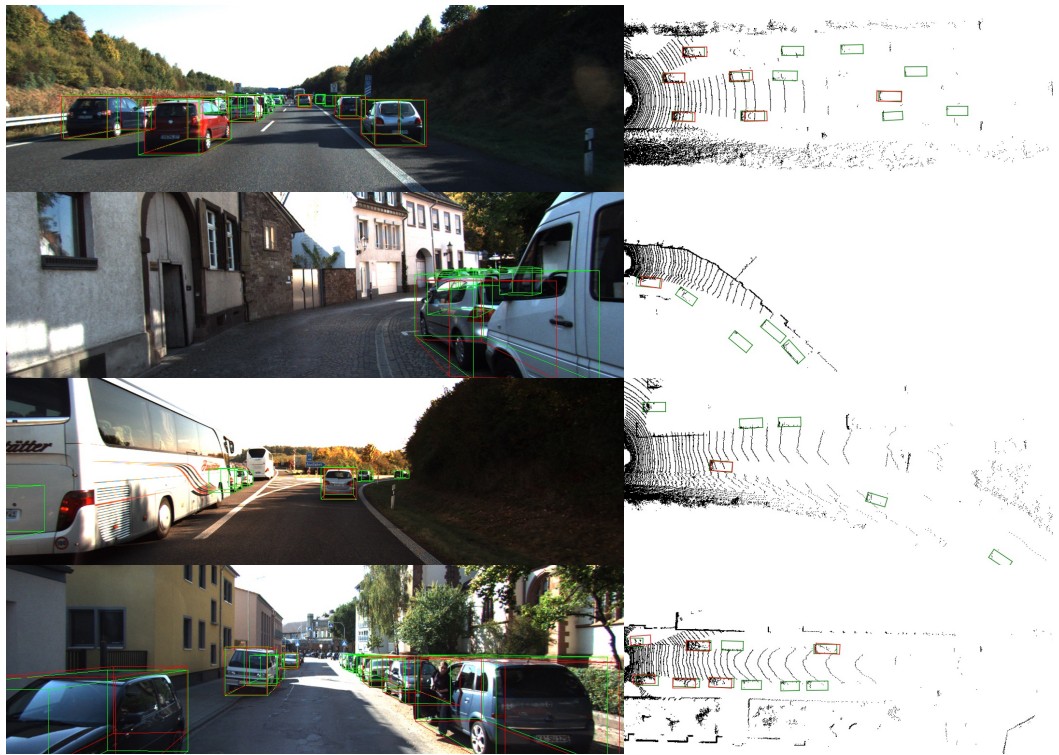

(a) Bad case in our methods. Low recall rate of the 2D detector is shown.

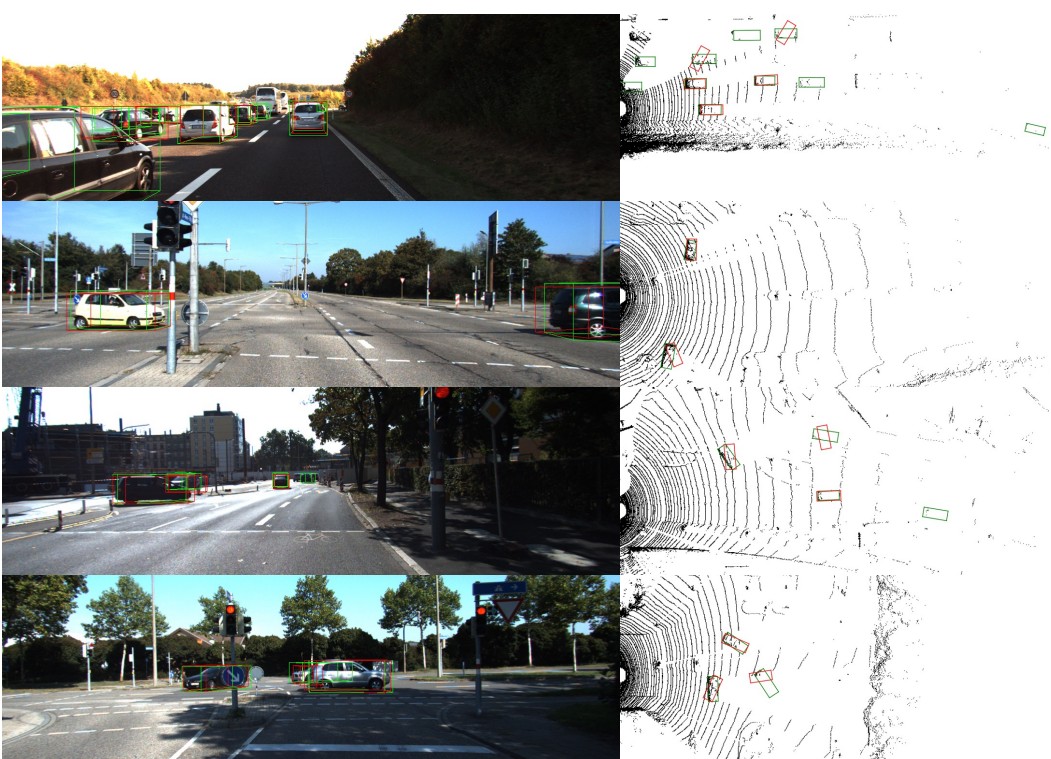

(b) Bad case in our methods. Rotation errors is obvious in these cases.

Figure 4: Bad cases on the KITTI validation dataset. The 3D ground-truth boxes and our predictions are drew in green and red, respectively. (a) We show the failure cases of the 2D detector in different scenarios caused by occlusions, small sizes, etc., which directly influence the accuracy of the 3D detector. (b) Large rotation errors are caused by appearance misperception.