# OpenReview forum: "Progressive Coordinate Transforms for Monocular 3D Object Detection"
_NeurIPS.cc/2021/Conference — NeurIPS 2021 Poster_

### Official Review · Reviewer_rVMM · 2021-07-10

**Rating:** 3
**Confidence:** 5

**Summary:**

This paper proposes progressive coordinate transforms to solve the problems in monolar 3D object detection. But there are some problems with workload and experiments.

**Limitations And Societal Impact:**

The authors have not addressed the limitations and potential negative societal impact of their work.

**Main Review:**

1. The main contribution of this paper focuses on the application of gradient enhancement to localization prediction. Global context encoding seems just a small improvement to former algorithms. Therefore, the overall contribution is not particularly large.

2. The experiment in this paper is carried out on the KITTI data set. The results on the test set in Table 6 should be obtained from the official website submitted to the official website, but the corresponding data (anonymous) is not seen on the KITTI website. So where did the experimental results in Table 6 come from? If it was obtained by KITTI, please submit it anonymously; if not, then there is a problem with the authenticity of the data. In addition, KITTI includes three types of detection object. Why does the experimental result only include the detection result of the car? How about the other two types of test results?

3. The experiment is only carried out on the KITTI dataset, but more experiments on data sets such as waymo and nuscences should be carried out. In addition, the method in this paper is actually a 3D object detection algorithm based on monocular images, so it should be compared with other algorithms of the same type (3D object detection based on monocular image), such as GrooMeD-NMS[1], DDMP-3D[2], CaDDN[3], etc, but this paper does not compare them.

[1] A. Kumar, G. Brazil and X. Liu: GrooMeD-NMS: Grouped Mathematically Differentiable NMS for Monocular 3D Object Detection. CVPR 2021.
[2] L. Wang, L. Du, X. Ye, Y. Fu, G. Guo, X. Xue, J. Feng and L. Zhang: Depth-conditioned Dynamic Message Propagation for Monocular 3D Object Detection. CVPR 2020.
[3] C. Reading, A. Harakeh, J. Chae and S. Waslander: Categorical Depth Distribution Network for Monocular 3D Object Detection. CVPR 2021.

4. This paper claims to design a lightweight coordinate-based network, so what is the amount of calculation and complexity of the algorithm in this paper? How does it compare to other coordinate-based networks? The comparison between the results of this part and other methods is also very important, but it does not appear.


**Time Spent Reviewing:**

8 hours

---

> ### Author Response · Authors · 2021-08-10
> **Response to Reviewer rVMM**
>
> We thank the reviewer for the detailed review as well as the suggestions for improvement. Our response to the reviewer’s comments is below:
>
> #### **Q1: Overall contribution is not particularly large.**
>
> Despite the techniques we propose are conceptually simple, they are lightweight, effective, and most importantly, built upon insightful observations. As agreed by the reviewer, simplicity is key.
> 1. Regarding the CLB module, we find that existing coordinate-based approaches usually overlook the localization problem which is the key to accurate 3D detection. Thus, we propose to progressively refine the representation via a boosting mechanism alongside the confidence score prediction to improve the localization accuracy. This method is new (i.e., we advocate to improve the input representation of coordinate-based methods), lightweight (i.e., only introduce marginal parameters, 3.41M compared with the base method PatchNet[1] 48.39M, see more in our response to Review3 Q4), and generalize well (i.e., we have shown improved performance over multiple base methods on different datasets).
> 2. Regarding the GCE module, it is the most straightforward way to incorporate semantic representation and is robust for different 2D detectors. We agree that it has small improvement with 2D detector in RTM3D[2] (Table 4 in the main paper), but it shows clear improvement with 2D detector in DDPM-3D[3] (Table 1 in the supp) under the same setting. We can see its comparison in the table below, where `Baseline` applies localization boosting without confidence constraint, and `G + GCE` is the pattern we explore GCE in our final PCT.
> | Method   | AP3D@Mod. |  AP3D@Easy | AP3D@Hard |
> | :--------: | :---------: | :----------: | :---------: |
> | Baseline |   21.92   |   32.66    |   19.86   |
> | G + GCE  |   25.66   |   36.17    |   23.34   |
>
> Overall, we believe our contribution is significant compared to prior work. We outperform or are competitive to most recent CVPR21/ICCV21 works in KITTI, for example, we are 5x lighter than DDMP-3D[3] (CVPR21) in terms of model parameters but achieves better performance (Ours: 13.37/21.00/11.31  vs DDMP-3D: 12.78/19.71/9.80). Our method is versatile, and can be used together with any coordinated-based method (e.g., Pseudo-LiDAR [4]) for improved performance.
>
> #### **Q2: The test results.**
>
> - Public results: We have now made our results public. Please refer to the “PCT” entry (around 250th place at `Car` category) in http://www.cvlibs.net/datasets/kitti/eval_object.php?obj_benchmark=3d
> Our detailed results are in http://www.cvlibs.net/datasets/kitti/eval_object_detail.php?&result=f72efc1108931352f096e16b4d1d9da175357223
> From the leaderboard, our proposed PCT outperforms most of the recent state-of-the-art methods including Ground-Aware MonoRCNN (ICCV 2021), DDMP-3D (CVPR 2021), GrooMeD-NMS (CVPR 2021), MonoRUn (CVPR 2021), monodle (CVPR 2021), YoloMono3D (ICRA 2021), Kinematic3D (ECCV 2020), D4LCN (CVPR 2020) and our baseline PatchNet (ECCV 2020).
> - Results on Pedestrian and Cyclist. Most existing works, especially coordinated-based methods, only report numbers on the car category (PatchNet, GrooMeD-NMS, Kinematic3D, etc.), so we follow the default protocol to compare with them. As suggested, we do have the results for the other two categories `Pedestrian/Cyclist` on KITTI validation set in the table below. From the table, we can see that our model still outperforms the baseline approach PatchNet in both `Pedestrian/Cyclist` categories. We will include these results in the revised version.
> | Method		|AP3D@Mod.		|  AP3D@Easy 	| AP3D@Hard 	|
> | :----------------------:	| :-----------------: 		| :--------------------: 	| :--------------------: 	|
> | PatchNet  		| 12.17/11.60		| 14.55/13.76		| 12.00/11.37		|
> | Our PCT		| 15.31/12.28		| 17.19/15.98		| 13.12/12.19		|
>
> #### **Q3: Experiments on other datasets.**
> Thank you for your suggestion. We have carried out experiments on Waymo Open dataset to verify the generalization of our proposed methods. We show the results (mAP/mAPH) in the table below, which includes base method PatchNet and our PCT under IoU=0.7 on Car category.
>
> | Method 	| Level 		| Overall |  0 - 30m | 30 - 50m | 50 - infinity|
> | :--------------:	| :--------------:	| :--------------: | :--------------: | :--------------: | :--------------: |
> | PatchNet	| LEVEL_1 	| 0.39/0.37 | 1.67/1.63 | 0.13/0.12 | 0.03/0.03 |
> | PCT  		| LEVEL_1	| 0.89/0.88 | 3.18/3.15 | 0.27/0.27 | 0.07/0.07 |
> | PatchNet	| LEVEL_2 	| 0.38/0.36 | 1.67/1.63 | 0.13/0.11 | 0.03/0.03 |
> | PCT  		| LEVEL_2	| 0.66/0.66 | 3.18/3.15 | 0.27/0.26 | 0.07/0.07 |
>
> From the table, we can see that PCT outperforms the PatchNet on mAP/mAPH consistently, which verifies the effectiveness of our method.
>
>
>
> Regarding comparison, we will include the comparison to recent state-of-the-art methods in the revised version. We also put the table here. As we can see, our PCT outperforms them in all settings, except a bit lower on the `Mod.` subset compared to CaDDN.
>
> | Method		| Conference |  AP3D@Mod.		|  AP3D@Easy 	| AP3D@Hard 	|
> | :----------------------:	| :-----------------: 		| :-----------------: 		| :--------------------: 	| :--------------------: 	|
> | GrooMeD-NMS[5]  	| CVPR2021	| 12.32		| 18.10		| 9.65 		|
> | DDMP-3D[3]		| CVPR2021	| 12.78		| 19.71		| 9.80		 |
> | CaDDN[6]		| CVPR2021	| 13.41		| 19.17		| 11.46		 |
> | PCT (Ours)	            | -		| 13.37		| 21.00		| 11.31		 |
>
> #### **Q4: How lightweight of the proposed method compare to other methods?**
> We appreciate the valuable feedback, we will add discussion on efficiency in the revised version. Here, we put the comparison in terms of model parameters and corresponding test results in the table below.
>
> |    Method	| Type		| Params 	| AP3D|
> | :-------------:	| :-----------------: | :-----------: | :--------------------------------:	|
> | CaDDN[6]			| Pixel		| 191.24M	|13.41 / 19.17 / 11.46 |
> | DDMP-3D[3]			| Pixel		| 285.50M	|12.78	/19.71 / 9.80	|
> | PatchNet[1]			| Coordinate	| 48.39M	| 11.12 / 15.68 / 10.17 |
> | PatchNet + PCT		| Coordinate	| 51.80M 	| 13.37 / 21.00 / 11.31 |
>
> From the table we can see that our proposed PCT only introduces 3.41M extra parameters, which is marginal compared to the base method PatchNet with 48.39M parameters. This verifies that our proposed PCT is lightweight, but can achieve better performance than PatchNet.
>
> Besides, we also compare our method to the model sizes of recent pixel-based methods, such as CaDDN and DDMP-3D. Our final model (PatchNet + PCT) is much smaller than pixel-based methods (5x lighter than DDMP-3D[3]) but achieves competitive performance, which demonstrates that coordinate-based methods are promising and effective.
>
>
> #### **References**
> [1] Xinzhu Ma, Shinan Liu, Zhiyi Xia, Hongwen Zhang, Xingyu Zeng, and Wanli Ouyang. Re-thinking pseudo-lidar representation. In ECCV, 2020.
>
> [2] Peixuan Li, Huaici Zhao, Pengfei Liu, and Feidao Cao. Rtm3d: Real-time monocular 3ddetection from object keypoints for autonomous driving. In ECCV, 2020.
>
> [3] Li Wang, Liang Du, Xiaoqing Ye, Yanwei Fu, Guodong Guo, Xiangyang Xue, Jianfeng Feng,and Li Zhang. Depth-conditioned dynamic message propagation for monocular 3d objectdetection. In CVPR, 2021.
>
> [4] Yan Wang, Wei-Lun Chao, Divyansh Garg, Bharath Hariharan, Mark Campbell, and Kilian QWeinberger. Pseudo-lidar from visual depth estimation: Bridging the gap in 3d object detectionfor autonomous driving. In CVPR, 2019.
>
> [5] A. Kumar, G. Brazil and X. Liu: GrooMeD-NMS: Grouped Mathematically Differentiable NMS for Monocular 3D Object Detection. CVPR 2021.
>
> [6] Cody Reading, Ali Harakeh, Julia Chae, and Steven L. Waslander. Categorical depth distribu-tionnetwork for monocular 3d object detection. InCVPR, 2021.

---

> ### Author Response · Authors · 2021-08-31
> **Request for feedback on the rebuttal**
>
> Dear Reviewer rVMM,
>
> We appreciate your time for reviewing, and we really want to have a further discussion with you to see if our response solves the concerns. We have addressed all the thoughtful questions raised by the reviewer (eg, contribution, test results, waymo results and lightweight model) and we hope that our work’s impact and results are better highlighted with our responses. It would be great if the reviewer can kindly check our responses and provide feedback with further questions/concerns (if any). We would be more than happy to address them. Thank you!
>
> Best wishes,
>
> Authors

---

> ### Author Response · Authors · 2021-09-02
> **Request for feedback on the rebuttal**
>
> Dear Reviewer rVMM,
>
> Thanks again for your valuable comments and suggestions. As the discussion phase is nearing its end, we wondered if you might still have any concerns that we could address. We believe our responses on *contribution, test results, waymo results and lightweight model* addressed all your questions/concerns, and hope our response helps your final recommendation. Thank you.
>
> Best wishes,
>
> Authors

---

### Official Review · Reviewer_gTZB · 2021-07-15

**Rating:** 8
**Confidence:** 4

**Summary:**

The paper first analyzes where the main performance loss is in the
inference of 3D bounding boxes from 2D images and concludes it is
in the 3D box center estimation phase. The introduced method
is an add-on iterative refinement which can boost performance of
several algorithms, and improves state-of-art when boosting
PatchNet, as applied to the KITTI dataset.


**Ethical Concerns:**

There are general ethical issues about the impact of autonomous vehicles.

**Limitations And Societal Impact:**

I saw no obvious weaknesses. I was curious about the magnitude of the
weights learned.
There was a slight confusion between T=3 and +2 in table 3. Presumably
they mean the same thing, so it might be better to use the T values
rather than the increments.


**Main Review:**

The paper introduces an iterative box center coordinate refinement stage
into the 3D BBox estimation (based on 2D box detections and monocular
depth estimates).
Confidence weights are estimated for each progressive refinement.
Semantic information is also added, which also boosts box location
estimation. An ablation study shows that each of the 3 added components
boosts performance a little. The proposed addition is also added to
the Pseudo-LiDAR algorithm, also improving performance.
Comparison of the number of iterations of the localization boost was
done to select the best number of iterations (3).

The paper is good: some nice analysis and ideas, good experiments, clearly written, but is also incremental; hence the proposed score of 8.

**Time Spent Reviewing:**

2

---

> ### Author Response · Authors · 2021-08-10
> **Response to Reviewer gTZB**
>
> We appreciate very much your constructive feedback and find our work with good ideas, analysis and experiments.
>
> - Regarding T=3 and +2 in Table 3. Yes, your interpretation is correct, they mean the same thing. We will correct Table 3 to make it consistent in the revised version.
>
> - I assume the magnitude of the weights learned you mean is the learned confidence weights. If this is the case, we can track the values of them at different epochs to understand it better. To be specific, we sample the confidence weights of a single image from the initial and last epoch; the changes of confidence weights and corresponding center losses at each iteration can be found in the table below.
> | epoch | 1th step (weight / loss) | 2th step (weight / loss) |  3th step (weight / loss) |
> | :--------: | :-------------------------------: | :-------------------------------: | :-------------------------------: |
> | init | 0.47 / [0.094 0.111 0.491] | 0.50 / [0.242 0.181 0.497 ] | 0.54 / [0.286  0.191 0.527] |
> | last | 1. / [0.005 0.097 0.259] | 1. / [0.149 0.120 0.215] / | 1. / [0.005 0.097 0.259] |
>
> From the Table, we can observe that the initial weights for each boosting iteration are around 0.5, while they are around 1 after optimization. This phenomenon confirms Equation (5) in the paper, which indicates that the network is more and more confident about the results they predict with stepwise refinement.

---

> ### Comment · Reviewer_gTZB · 2021-08-27
> **Comments on Author Reply**
>
> I think that the authors have adequately addressed the issues raised by the reviewers. I still support an accept, although it is not a top paper.

---

### Official Review · Reviewer_vnDo · 2021-07-16

**Rating:** 6
**Confidence:** 4

**Summary:**

This paper presents a "progressive coordinate transform" strategy for improving monocular 3D object detection. Specifically, the authors identify that the main bottleneck in performance for monocular 3D object detection is the localization performance and tailor a solution towards progressively refining the coordinate proposals with multiple stacked modules before making the final 3D bbox prediction. The authors also propose a "Global context encoding" which processes and concatenates the visual features from the object proposal before the final 3D bbox prediction. This empirically results in an improvement in overall 3D bbox detection performance - achieving state of the art of the KITTI monocular 3D object detection benchmark. The primary contribution of this paper is identifying localization performance as a key bottleneck for 3D monocular object detection and tailoring a simple solution to improve performance over prior work.

**Limitations And Societal Impact:**

The authors are encouraged to comment on the technical novelty and significance of improvements over prior work on the KITTI dataset.

**Main Review:**

*Strengths*
-  Table 1 in the paper is most insightful in my opinion and identifies localization as a bottleneck (vs size or rotation prediction). This is also somewhat expected because the variation in dimensions for cars is not expected to be high and orientation is also expected to be in a constrained range in driving scenarios. However quantifying this still has value. The authors build from this simple intuition and propose an iterative refinement method after the coordinate proposal module in standard 3D monocular object detection methods. The simplicity is key.

*Weaknesses*
- The two major concerns I have with the paper are technical novelty and significance of improvement over prior work. The major contribution in this work is really the "CLB" module which is primarily an iterative refinement module which predicts residuals between the ground truth and the previous prediction at each stage. The GCE module which appends a visual feature vector computed from the detector backbone isnt necessarily novel and has relatively low effect on performance (<0.4AP in table 4).

- The final improvement on the validation set seems significant for the "Hard" subset (~3.5AP in AP3D) but might not be very significant in my opinion on the other subsets and metrics. This is true when comparing against non coordinate-based methods on the KITTI test set in Table 6.

- The figures in the paper often do not add much to the discussion. Fig 1 lacks more visuals and primarily a set of blocks. The caption for Fig 2(a) does not do a good job of explaining it and one has to follow the text very closely to understand the schematic here. The top row if FIg 3(a) is also hard to follow. The authors mention "peakier" red distribution curves indicating decreasing localization error over iterations but this is not readily visible to me. The BEV visualization and image bbox visualization in Fig 4 are also very small and the boxes themselves are hard to see. Overall, the figures in the manuscript need quite some improvement.



**Time Spent Reviewing:**

2

---

> ### Author Response · Authors · 2021-08-10
> **Response to Reviewer vnDo**
>
> We thank the reviewer for the detailed review as well as the suggestions for improvement. Our response to the reviewer’s comments is below:
> #### **Q1:  Technical novelty and significance of improvement over prior work.**
> Despite the techniques we propose are conceptually simple, they are lightweight, effective, and most importantly, built upon insightful observations. As agreed by the reviewer, simplicity is key.
> 1. Regarding the CLB module, we find that existing coordinate-based approaches usually overlook the localization problem which is the key to accurate 3D detection. Thus, we propose to progressively refine the representation via a boosting mechanism alongside the confidence score prediction to improve the localization accuracy. This method is new (i.e., we advocate to improve the input representation of coordinate-based methods), lightweight (i.e., only introduce marginal parameters, 3.41M compared with the base method PatchNet[1] 48.39M, see more in our response to Review3 Q4), and generalize well (i.e., we have shown improved performance over multiple base methods on different datasets).
> 2. Regarding the GCE module, it is the most straightforward way to incorporate semantic representation and is robust for different 2D detectors. We agree that it has small improvement with 2D detector in RTM3D[2] (Table 4 in the main paper), but it shows clear improvement with 2D detector in DDPM-3D[3] (Table 1 in the supp) under the same setting. We can see its comparison in the table below, where `Baseline` applies localization boosting without confidence constraint, and `G + GCE` is the pattern we explore GCE in our final PCT.
> | Method   | AP3D@Mod. |  AP3D@Easy | AP3D@Hard |
> | :--------: | :---------: | :----------: | :---------: |
> | Baseline |   21.92   |   32.66    |   19.86   |
> | G + GCE  |   25.66   |   36.17    |   23.34   |
>
> Overall, we believe our contribution is significant compared to prior work. We outperform or are competitive to most recent CVPR21/ICCV21 works in KITTI, for example, we are 5x lighter than DDMP-3D[3] (CVPR21) in terms of model parameters but achieves better performance (Ours: 13.37/21.00/11.31  vs DDMP-3D: 12.78/19.71/9.80). Our method is versatile, and can be used together with any coordinated-based method (e.g., Pseudo-LiDAR [4]) for improved performance.
>
>
> #### **Q2: Significant improvement on the `Hard` subset but not very significant on the other subsets.**
> We want to emphasize that the significant improvement on the `hard`  subset perfectly justifies the effectiveness of our CLB module. For the KITTI dataset, the `hard` subset usually consists of occluded or small objects, which is very challenging to localize them correctly. Our progressive refinement strategy helps to alleviate the localization difficulty problem, thus improves the most on the `hard` subset. Besides, our results on the test set of KITTI also outperform the base method PatchNet, by 2.25%/5.32%/1.14\% on Mod./Easy/Hard settings, respectively. This indicates that CLB is not tuned towards any specific subset, but can generalize.
>
>
>
> #### **Q3: The figures in the paper often do not add much to the discussion.**
> Thank you for the suggestions, we will improve the figures in the revised version.
>
> - Fig1: Our goal is to visualize the workflow of generic coordinate-based methods, so we use blocks to indicate the modeling process. We will add more visuals to make it more readable.
> - Fig2: We will add more details from the text to improve the caption.
> - Fig3(a): We attempt to visualize the distribution of localization errors on KITTI validation set. We compute the x/y/z errors compared with ground truth, so they are more accurate if the errors tend to be zero. Hence, when the curve is more thin, tall, and closer to zeros, the localization is more accurate. Therefore, we can see from Figure 3(a) that the red curve achieves the best localization performance which meets our expectation. We will put these insights in the caption of Fig3 to make it clear.
> - Fig4: We follow the same visualization format from literature (DDMP-3D[3], CVPR21), but we can rearrange the layout (i.e., make them into 4 horizontal rows) to make the bbox look larger.
>
> #### **References**
> [1] Xinzhu Ma, Shinan Liu, Zhiyi Xia, Hongwen Zhang, Xingyu Zeng, and Wanli Ouyang.  Re-thinking pseudo-lidar representation. In ECCV, 2020.
>
> [2] Peixuan Li, Huaici Zhao, Pengfei Liu, and Feidao Cao.   Rtm3d:  Real-time monocular 3ddetection from object keypoints for autonomous driving. In ECCV, 2020.
>
> [3] Li Wang, Liang Du, Xiaoqing Ye, Yanwei Fu, Guodong Guo, Xiangyang Xue, Jianfeng Feng,and Li Zhang.   Depth-conditioned dynamic message propagation for monocular 3d objectdetection. In CVPR, 2021.
>
> [4] Yan Wang, Wei-Lun Chao, Divyansh Garg, Bharath Hariharan, Mark Campbell, and Kilian QWeinberger. Pseudo-lidar from visual depth estimation: Bridging the gap in 3d object detectionfor autonomous driving. In CVPR, 2019.

---

> ### Author Response · Authors · 2021-08-31
> **Request for feedback on the rebuttal**
>
> Dear Reviewer vnDo,
>
> We appreciate your time for reviewing, and we really want to have a further discussion with you to see if our response solves the concerns. We have addressed all the thoughtful questions raised by the reviewer (eg,  novelty and performance) and we hope that our work’s impact and results are better highlighted with our responses. It would be great if the reviewer can kindly check our responses and provide feedback with further questions/concerns (if any). We would be more than happy to address them. Thank you!
>
> Best wishes,
>
> Authors

---

> ### Comment · Reviewer_vnDo · 2021-08-31
> **Updating recommendation after rebuttal**
>
> I would like to thank the authors for their detailed rebuttal and updated comparisons to more recent monocular 3D object detection works in the response to Reviewer rVMM. While I still believe that technical novelty might not be the paper's strong suit here, I appreciate the authors' dedication to thorough quantitative evaluations (e.g. on the Waymo Open dataset) and demonstrating the efficacy of such a simple procedure on 3D object detection performance. If accepted, I would also like to see the updates made to the figures as indicated by the authors in the rebuttal and better captions for the same.

---

### Decision · Program_Chairs · 2021-09-27

**Decision:**

Accept (Poster)

**Comment:**

At the time of rebuttal, the reviewers had widely varying opinions on the paper. Major concerns included general clarity as well as a set of fairly concrete experimental issues. The authors responded to the reviews with detailed new experiments to address the reviewer concerns. Reviewer vnDo was persuaded by these experiments and raised their score to an acceptance. Reviewer rVMM did not participate in post-rebuttal discussion. The AC examined rVMM's requests and the authors' responses. The AC cannot speak on behalf of rVMM; however, in the AC's view, much of the reviewers' concerns are ostensibly addressed directly with reasonable experiments. On balance, given that the consensus of the active reviewers is acceptance, and the rebuttal seems likely to address the remaining reviewer's concerns, the AC is strongly inclined to recommend acceptance.

The AC would suggest that the authors:
- Incorporate, to the best of their ability, the results presented in the rebuttal. If they do not fit, the authors should report them in the supplement.
- Take Reviewer vnDO's comments on figures and clarity seriously, and update them and their captions.
- As noted by the reviewers, the authors should extend their discussion of social impacts.

Both promises/new results were instrumental in the paper's acceptance, and the authors will maximize the impact of their paper if they make these changes.